# The Effects of Land-Use and Climatic Changes on the Hydrological Environment in the Qinling Mountains of Shaanxi Province

**Kuifeng Zhao** [1], **Jing Li** [2,*], **Xinping Ma** [2,3] and **Chenhui Deng** [2,3]

1   Shaanxi Provincial Meteorological Bureau, Xi'an 710014, China
2   School of Geographic Science and Tourism, Shaanxi Normal University, Xi'an 710119, China
3   School of Geography and Environment, Xianyang Normal University, Xi'an 712000, China
*   Correspondence: lijing@snnu.edu.cn; Tel.: +86-137-2061-8191

**Abstract:** Under the dual influence of climate change and land-use change, different protection policies in Qinling Mountains lead to different hydrological responses. On the basis of land-use and climatic changes in the Qinling Mountains of Shaanxi province, we constructed a response model among land use, climate, and runoff in the Qinling Mountains through the Soil and Water Assessment Tool (SWAT) model. The Patch-generating Land Use Simulation (PLUS) model was used to predict and simulate land-use types of the Qinling Mountains in 2025 and 2030. On the basis of the current ecological protection policy in China, two scenarios of land use were set up to find the best method for forestland protection. The results show the following. The SWAT model is applicable to the Weihe River and Hanjiang River basins of the Qinling Mountains, the simulation results were verified, and the Nash coefficient was above 0.6. Under future climate change and land-use patterns, runoff in the Qinling Mountains watershed shows an upward trend, and the runoff in the Hanjiang River basin increased by more than that in the Weihe River basin, with change rates of 47.471 and 33.356 $m^3/10a$, respectively. According to the future trend of the two different scenarios, the increase degree of runoff in the natural scenario of Weihe River basin was 16.567 $m^3/10a$ higher than that in the woodland scenario, and the increase degree of runoff in the Hanjiang River basin was 17.692 $m^3/10a$ higher than that in the woodland scenario. Therefore, blindly increasing the forestland area in Qinling Mountains cannot achieve a better hydrological effect.

**Keywords:** Qinling Mountains; climate change; land use; hydrological effects; SWAT model; PLUS model





## 1. Introduction

The Qinling Mountains are an important north–south natural environment dividing line in China and an area with rich biodiversity [1]. In recent years, with the Chinese government's enhanced support for ecological environment protection, a series of ecological projects have been implemented in the Qinling Mountains, such as returning farmland to forests and natural forest protection, which greatly changed land use and cover in this area. Research on climate change in the Qinling Mountains mainly focuses on the response of vegetation and phenology to climate factors [2–5] or simply on the spatiotemporal characteristics of climatic factors of the Qinling Mountains [6,7]. Previous studies found that vegetation in the Qinling Mountains is highly sensitive to climate change. In recent years, climate change in the Qinling Mountains has also shown a special temporal and spatial change law with the intensification of global climate change [8–10]. The Qinling Mountains, with their special geographical location, great height, and well-developed forest ecosystem, are a key area for the development of the national economy and scientific research. However, under the influence of climate change, the extreme temperature in Qinling Mountains increased in the past 55 years, the 0 °C isotherm obviously moved

northward and increased [11], and the phenology of mountain vegetation also changed significantly [8,12–14], especially in high-altitude areas, which are more sensitive to climate change [15,16]. At the same time, in the context of climate change, with the development of the economic era, human activity has indirectly affected the climate and natural environment of the Qinling Mountains by changing the types of land use. In recent years, with the continuous strengthening of ecological protection in the Qinling Mountains, the implementation of ecological land-use policies has gradually affected the natural environment of the Qinling Mountains. In the context of the current implementation of environmental protection policies, what kind of response mechanism exists among land-use, climate-change, and natural factors in the Qinling Mountains? The project focuses on this issue, aiming to examine the response relationship between land-use and climatic changes in the Qinling Mountains, and provide a scientific basis for the rational adjustment of future land-use policies.

Research on the Qinling Mountains in China and abroad is summarized below. Most of the research on the Qinling Mountains can be divided into physical geographical factors, boundaries [17], ecosystem services [18] and quality [19], the response of vegetation and climate change [20], and land-use change and its environmental effects [21]. Although existing studies produced more indepth and clear results in the aspects of geographical factors, ecology, vegetation, and climate change, there are still few studies on the mutual response of land use, climate change, and basin hydrology in the Qinling Mountains. In particular, there are relatively few studies on land-use change in the Qinling Mountains, and more studies focused on precipitation, drought, temperature, snowfall, evaporation, water production, ecological quality, vegetation [22], and phenology [9]. However, in practice, land use, climate change, and hydrological changes are factors that interact with each other. On this basis, we propose a response relationship among land use, climate change, and the hydrological response effect in the Qinling Mountains. The SWAT model was used to simulate the response relationship among climate, land-use, and runoff in the Qinling Mountains, and the latest land-use model, PLUS, was used to explore the impact of different land-use development trends on the runoff environment under future climate change in order to provide a scientific basis for the policy guidance of land-use planning and optimize land use in the Qinling Mountains, and to mitigate the impact of climate change on future human activities and minimize property damage.

SWAT was developed by Dr. Jeff Arnold of the agricultural research center of the United States Department of Agriculture (USDA) in 1994. It is a semidistributed model that has been rapidly developed and applied in recent years. It mainly uses spatial information provided by remote sensing and geographic information systems to simulate a variety of different hydrological physical and chemical processes, such as water quantity and quality, and the transportation and transformation of pesticides [23]. The model has been widely used and developed for a long time [24], and its simulation effect is good [25]. SWAT was applied to simulate the relationship among runoff, land-use change, and climate change [26,27]. The soil and water assessment tool (SWAT) is a powerful modeling tool that can simulate many physical processes in the water cycle [28,29]. A large number of studies proved that distributed hydrological model SWAT can be better applied to the simulation of the impact of climate change on watershed water resources [30,31]. In this study, the SWAT model was used to simulate the runoff of the upper Hanjiang River basin from 2000 to 2015, the sufi-2 algorithm was used for an iterative operation to determine the optimal value of parameters, and the model was calibrated and verified. On this basis, the impact of climate change and human activities on the upper reaches of the Hanjiang River basin is studied.

In the past, the CA Markov model was widely used in the simulation research of land-use change. However, existing CA models have some shortcomings in both the mining strategy of transformation rules and the simulation strategy of dynamic landscape changes; most existing CA models also pay too much attention to the improvement of simulation technology and the correction of transformation rules. Therefore, for decision

makers, (1) existing CA models often play a relatively limited role in exploring the causes of land-use change [32], and (2) it is difficult to dynamically simulate patch-level changes in various land-use types, especially for forestland, grassland, and other natural land-use types [33]. Other land decision-making models either pay little attention to patch changes in natural land-use types or lack a flexible mechanism to deal with the patch changes of multiple types of land use [34]. Therefore, on the basis of grid data, some scholars proposed a patch generation land-use change simulation model that applies a new analytical strategy to better mine the incentives of various land-use changes. At the same time, the model includes a new multiclass seed growth mechanism that can better simulate the patch-level changes of multiclass land use [35]. Lastly, the model was coupled with the multiobjective optimization algorithm, and the simulation results could better support planning policies to achieve sustainable development [36,37].

On the basis of the above analysis, the research objectives of this paper are as follows: (1) the response relationship between land-use and climatic changes in the Qinling Mountains. (2) The forecast of future land-use and climatic changes in the Qinling Mountains. (3) The SWAT model-based construction of the response process among land use, climate change, and watershed runoff in the Qinling Mountains. (4) The analysis of future hydrological effects of land-use and climate change in the Qinling Mountains in combination with protection policies, so as to demonstrate reasonable protection policy recommendations.

## 2. Materials and Methods

### 2.1. Study Area

The study area mainly covers the hinterland of the Qinling Mountains in Shaanxi. It is bounded by the administrative unit near the ridge line in the north, some areas of Hanjiang River in the south, and the boundary of Shaanxi province in the east and west (Figure 1). The geographical coordinates range from $105°42'44''$–$111°3'29''$ E, $32°28'31''$–$34°40'59''$ N, with a total area of about 61,726.83 km$^2$. The obvious characteristics of this area are that the northern and southern slopes are quite different, and the intensity of human activity has obvious spatial differences [38]. The ecological environment of the Qinling Mountains' vegetation and watershed is affected by climate change and human activities. The influence of climate change in the Qinling mountains has a certain change rule in the horizontal and vertical directions [39].

### 2.2. Soil and Water Assessment Tool (SWAT) Model

The watershed hydrological process simulated by SWAT was divided into the land stage of the hydrological cycle (i.e., runoff generation and slope confluence) and the confluence stage of the hydrological cycle (i.e., river confluence). The former controls the input of water, sand, nutrients, and chemicals in the main river channel in each subwatershed. The latter determines the transportation of water, sand, and other substances from the river network to the outlet of the basin. The whole water circulation system follows the law of water balance [40].

$$SW_t = SW_0 + \sum_{i=1}^{t} (R_{day} - Q_{surf} - E_a - W_{seep} - Q_{gw})$$

where $SW_t$ is the final water content of soil, mm. $SW_0$ is the initial soil water content on day $i$, mm; $t$ represents time, $d$; $R_{day}$ refers to the precipitation on day $i$, mm; $Q_{surf}$ refers to the surface runoff on day $i$, mm; $E_a$ is the evapotranspiration of day $i$, mm; $W_{seep}$ refers to the amount of water entering the aeration zone from the soil profile on day $i$, mm; $Q_{gw}$ refers to the amount of return flow on day $i$, mm.

The sub-basins and hydrological response units (HRUs) were divided. According to the actual situation of the basin, the upper reaches of the Hanjiang River basin were divided into 109 sub-basins (Figure 2a) and 5051 HRUs. The Weihe River basin in Shaanxi province

was divided into 53 sub-basins (Figure 3a) and 2352 HRUs. Second, the soil database was constructed according to the soil types in the study area.

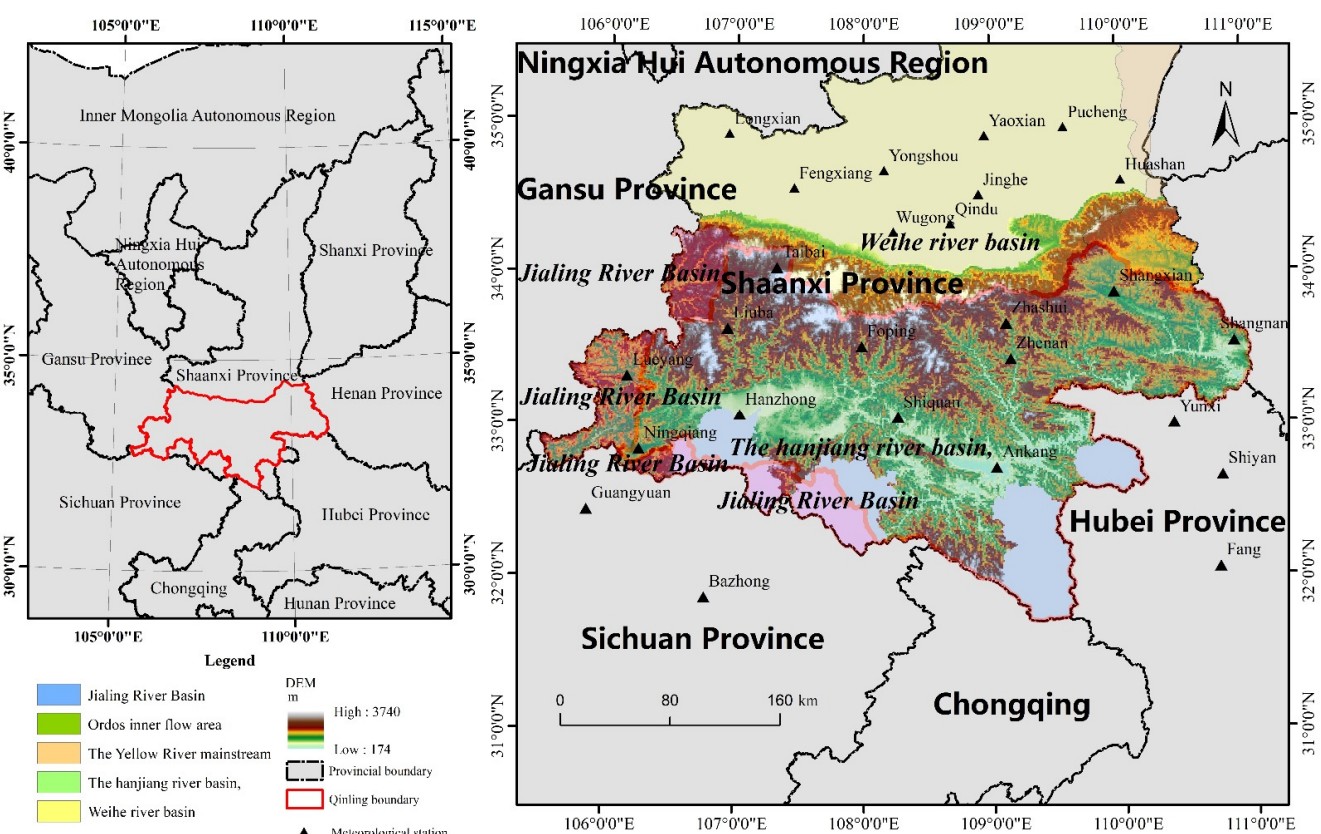

**Figure 1.** Overview of the study area.

### 2.3. PLUS Model

This paper adopts a CA model based on multiclass random patch seeds, that is, the cars module of the PLUS model. This module combines random-seed generation and a threshold decreasing mechanism, so that the PLUS model can automatically generate spatiotemporal dynamic simulation patches under the constraint of development probability. The mathematical principle of multiclass random-seed generation is as follows [41]:

$$OP_{i,k}^{1,t} = \begin{cases} P_{i,k}^1 \times (r \times \mu_k) \times D_k^t & if\ \Omega_{i,k}^t = 0\ and\ r\ < P_{i,k}^1 \\ P_{i,k}^1 \times \Omega_{i,k}^t \times D_k^t & all\ others \end{cases}$$

$$\Omega_{i,k}^t = \frac{con(c_i^{t-1}=k)}{n \times n-1} \times w_k$$

$$D_k^t = \begin{cases} D_k^{t-1} & if\ \left|G_k^{t-1}\right| \le \left|G_k^{t=2}\right| \\ D_k^{t-1} \times \frac{G_k^{t-2}}{G_k^{t-1}} & if\ 0 > G_k^{t-2} > G_k^{t-1} \\ D_k^{t-1} \times \frac{G_k^{t-1}}{G_k^{t-2}} & if\ G_k^{t-1} > G_k^{t-2} > 0 \end{cases}$$

The mathematical model of the threshold decreasing mechanism is as follows:

$$If\ \sum_{k=1}^N \left|G_c^{t-1}\right| - \sum_{k=1}^N \left|G_c^t\right| < Step\ Then,\ d = d+1$$

$$\begin{cases} Change & P_{i,c}^1 > \tau & and\ TM_{k,c} = 1 \\ Unchange & P_{i,c}^1 \le \tau & or\ TM_{k,c} = 0 \end{cases}$$

### 2.4. BCC/RCG-WG Weather Generator

Future daily meteorological data in this study were predicted by using China weather generator BCC/RCG-WG, developed by Dr. Liao Yaoming [42–44]. Through the statistical test of the simulation effect of BCC/RCG-WG in China, the results show that the annual average maximal and minimal temperature, sunshine hours, daily temperature range, monthly average maximal and minimal temperature, sunshine hours, and daily temperature range are very close to the measured results [45,46]. The BCC/RCG-WG weather generator could better simulate the daily maximal and minimal temperature, sunshine hours, and other nonprecipitation variables across China.

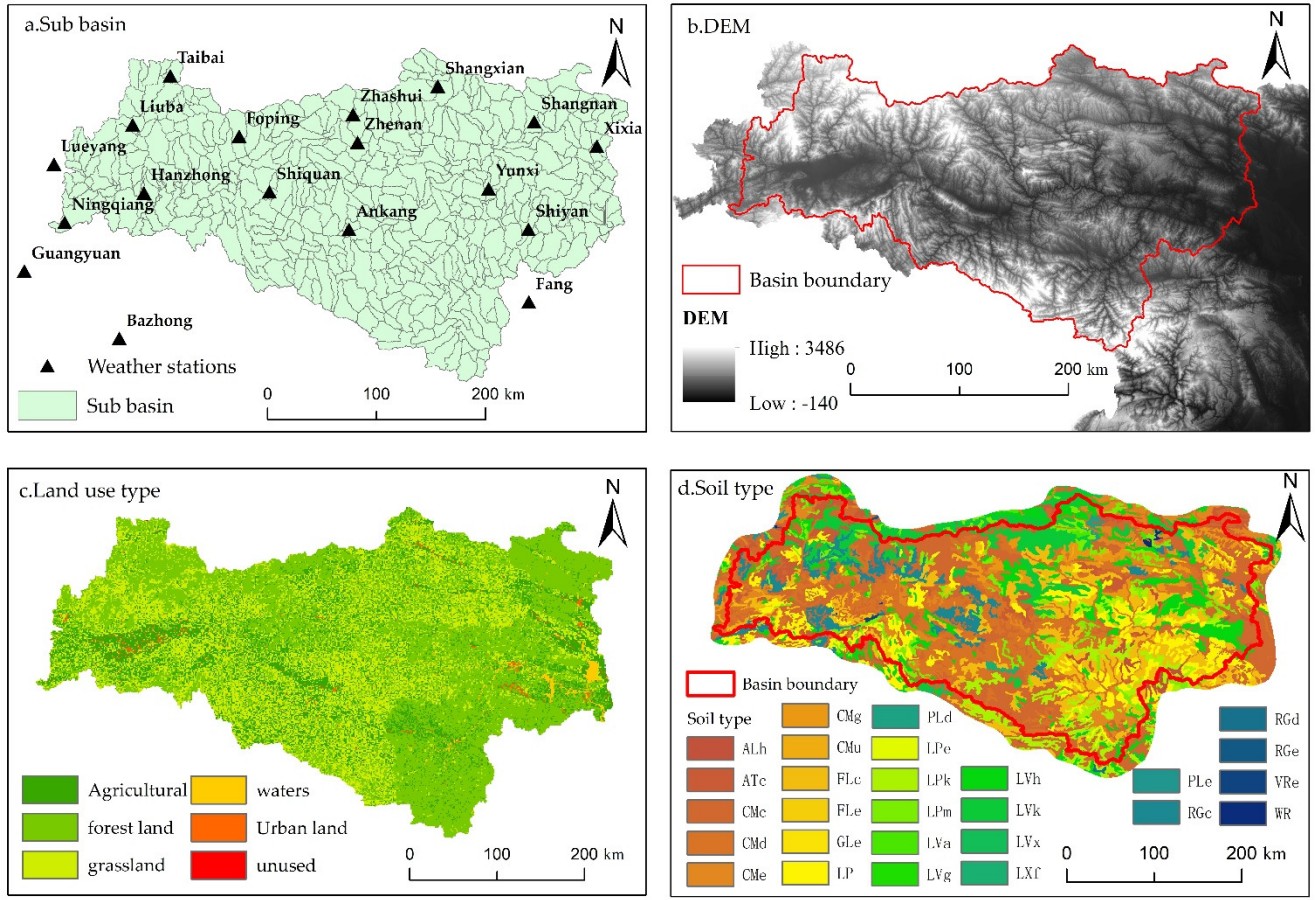

**Figure 2.** Spatial data required for SWAT model construction in the upper reaches of the Hanjiang River basin.

### 2.5. Scenario Setting

This paper adopts the natural land-use development and forestland-growth scenarios under the future climate conditions, and uses the BCC/RCG-WG weather generator to predict daily climatic data for the next 10 years. The natural land-use development scenario was land-use data for 2025 and 2030, predicted by the Markov chain method in the PLUS model. The model considers limiting and influencing factors, and can simulate and predict future land-use types. Therefore, the simulation results of this model were taken as the natural land-use development scenario. The policy-oriented land-use development scenario combines the current forestland protection policies. According to the growth index of the forestland area in the national forestland protection and utilization planning outline (2010–2020), a large number of forestland areas exceeding the index are artificially increased as forestland grows (Table 1).

**Table 1.** Scenario analysis settings.

| Climate Scenario | Land Use Scenario |
| --- | --- |
| Prediction of future climatic data based on BCC/RCG-WG weather generator | Natural development scenario (PLUS model) Forestland growth scenario (forestland protection policy) |

The scenario setting of the project was practically combined with the actual local situation, and the domestic climatic prediction model was adopted for future climatic predictions. The land-use scenario model also combines local influencing factors and policy guidance, so the research results are practical and referential.

*2.6. Data Sources*

The data used in the study are mainly the DEM of the Qinling Mountains; soil type and attributes; LUCC data of Qinling Mountains in 2005, 2010 and 2015; the daily temperature, precipitation, and wind-speed data of 13 meteorological stations in and around the study area; the monthly runoff data of Ankang station from 1970 to 2015; the daily runoff data of Huaxian and Yangxian stations in 2019 and 2020; a database of soil type maps and soil properties; the transportation network, residential areas, GDP, population data of the study area. Transportation-network, residential-area, GDP, and population data of the study area were provided by the national basic geographic information database (http://www.ngcc.cn/ngcc/html/1/391/392/16114.html (accessed on 21 January 2021)). All spatial data projection coordinates were unified to be WGS_1984_UTM_Zone_49n; the numbers of the rows and columns of pixels were also unified to be 548 and 339, respectively.

The data needed to build the SWAT model are shown in Table 2. The spatial data include meteorological-station information (longitude and latitude, altitude, etc.), watershed DEM (90 m; due to the scale of the study area and related research results [47], a DEM with a resolution of 90 m was adopted here), the land-use type map within the watershed, and the soil-type map with complete watershed coverage. Attribute data include daily maximal and minimal temperature, and daily precipitation data of the meteorological stations (.txt format), a meteorological database (wgen_user), and a soil database (usersoil). Land-use/-cover change (LUCC) data were obtained from the Data Center for Resources and Environmental Sciences, Chinese Academy of Sciences (http://www.dsac.cn/ (accessed on 15 February 2022)) at a resolution of 30 m. Landsat TM/ETM/OLI remote-sensing images were used as the main data source. Through image fusion, geometric correction, and image enhancement and mosaics, the human–computer interaction visual interpretation method was used to classify the land-use types in China into six first-level categories, namely, forestland, grassland, water source, residential land, and unused land.

**Table 2.** Attribute data required for SWAT model construction and their sources.

| Data Type | Data Sources |
| --- | --- |
| Digital elevation model (DEM) Spatial resolution, 90 m | Geospatial data cloud platform (http://www.gscloud.cn/ (accessed on 15 February 2022)) |
| Soil type map, 1000 m | World Soil Database (HWSD) |
| Soil property database | Chinese soil database, SPAW software |
| 2010/2015/2020 LUCC (1000 m) | Data Center for Resources and Environmental Sciences, Chinese Academy of Sciences (http://www.dsac.cn/ (accessed on 15 February 2022)) |
| Daily meteorological data of temperature and precipitation | China Meteorological Data Sharing Center (http://data.cma.cn/ (accessed on 20 February 2022)). |
| Hydrological data | Shaanxi Meteorological Bureau |

The data sources required for the PLUS model are shown in Table 3. We used the land-use data of the Qinling Mountains in 2010, 2015, and 2020 (Figure 4). The spatial resolution of the data was 1000 m. The data were from the 2020 version of global land-cover

data released by the Ministry of Natural Resources on 15 September 2020. Transportation-network, residential-area, GDP, and population data are provided by the national basic geographic information database (Table 3).

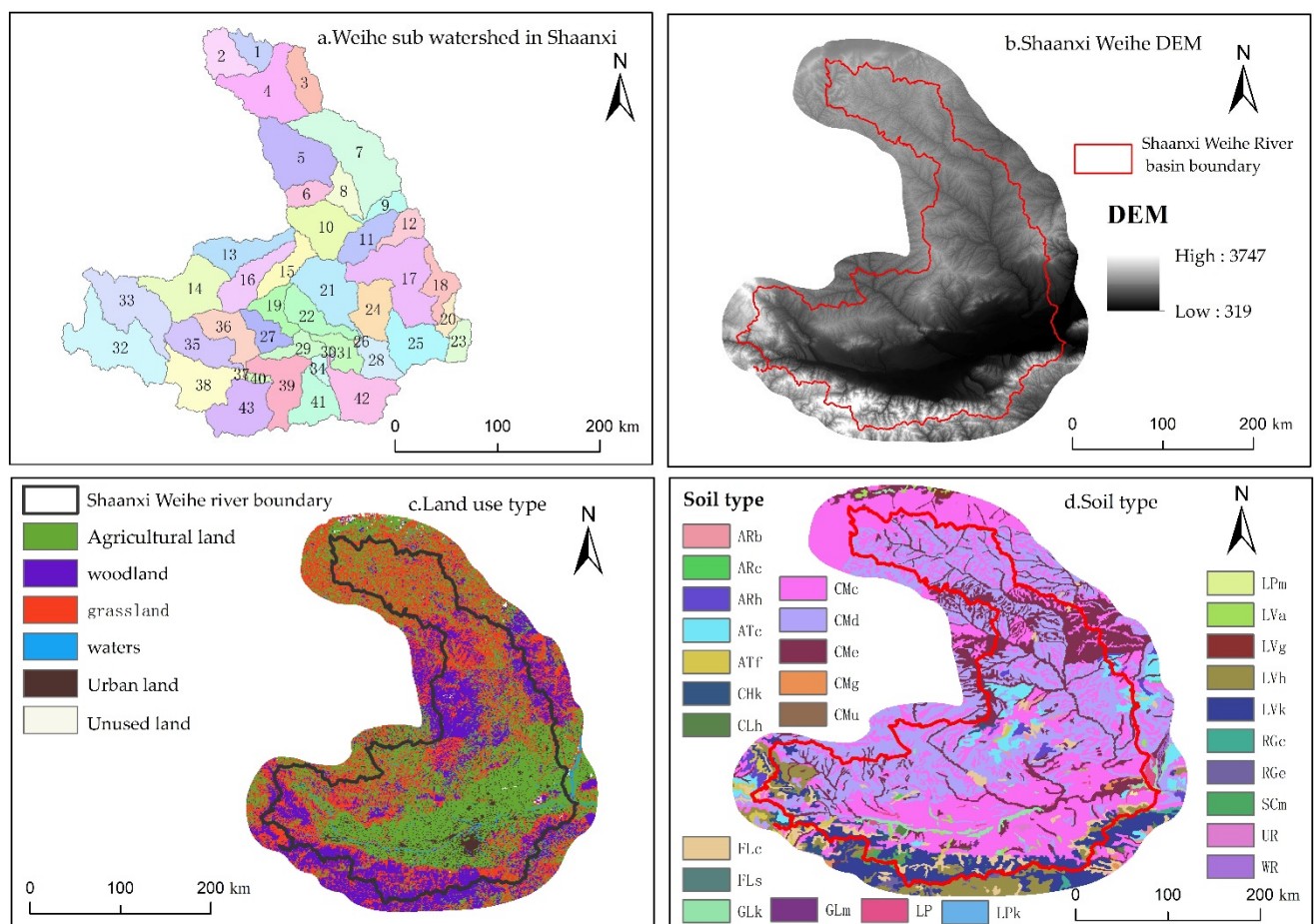

**Figure 3.** Spatial data required for SWAT Model construction in Shaanxi Weihe River basin.

**Table 3.** PLUS model data preparation.

| Type | Data | Description |
|---|---|---|
| Land-use data | Classification data of land use in Qinling Mountains in 2010 Classification data of land use in Qinling Mountains in 2015 Classification data of land use in Qinling Mountains in 2020 | (1) agricultural land, (2) forestland, (3) grassland, (4) water area, (5) construction land, (6) unused land. |
| Restricted conversion zone data | Constraints on land use | Water area (raster data) |
| Socioeconomic data | Population spatial data Spatial GDP data | http://www.geodoi.ac.cn/WebCn/Default.aspx/ (accessed on 21 January 2021) |
| | Distance to the transportation network Distance from settlements | National basic geographic information database |
| Environmental data | Elevationslope | Geospatial data cloud platform |

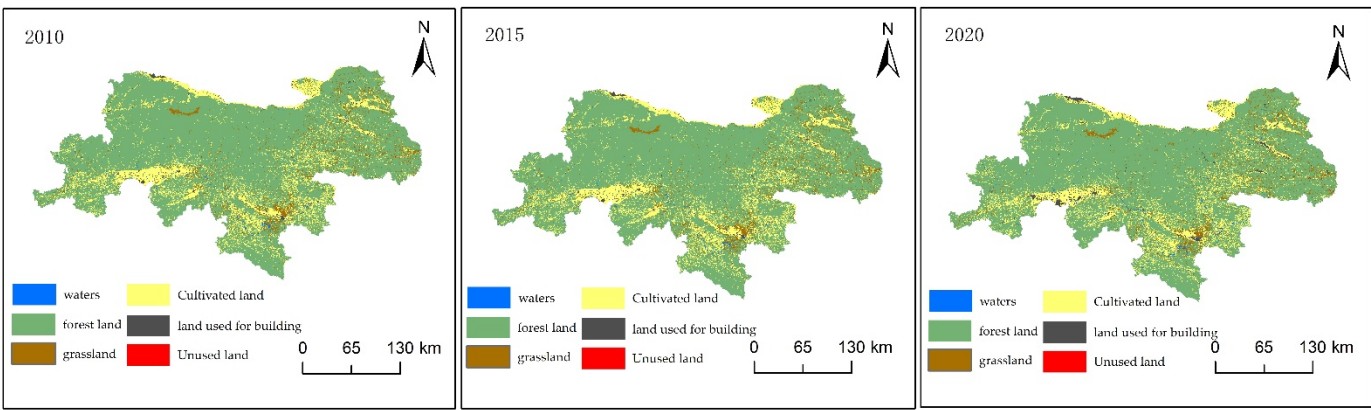

**Figure 4.** Land-use types of Qinling Mountains in 2010, 2015, and 2020.

## 3. Results

*3.1. Response Model Construction of Land Use, Climate Change, and Runoff Change in the Qinling Mountains*

Through the daily runoff data of the Yangxian and Huaxian hydrological stations in 2020, the project calibrated and verified the constructed SWAT models of Hanjiang and Weihe River basins in Shaanxi province.

(1)　Sensitivity parameter setting

Seven sensitivity parameters related to runoff simulation were selected in the study: runoff curve, base flow regression coefficient, base water-flow level threshold, the Manning coefficient of the main river channel, the effective hydraulic conductivity of the main river channel, the lag coefficient of the surface runoff, and the operational time of soil flow (Table 4). After several calibrations, the final calibration value was determined, as shown in Table 4. The calibration parameters were integrated into the model database and run again to obtain the final result.

**Table 4.** Selection and setting of sensitivity parameters.

| Serial Number | Parameter Name | Parameter Definition | Minimal Value | Maximal Value | Rate Constant Value |
|---|---|---|---|---|---|
| 1 | r__CN2.mgt | Runoff curve | −0.2 | 0.2 | −0.08 |
| 2 | v__ALPHA_BF.gw | Base flow regression coefficient | 0.0 | 1.0 | 0.7 |
| 4 | v__GWQMN.gw | Threshold of base flow level | 0.0 | 2.0 | 1.0 |
| 5 | v__CH_N2.rte | Main channel Manning coefficient | 0.0 | 0.3 | 0.19 |
| 6 | v__CH_K2.rte | Effective water conductivity of main channel | 0 | 3500 | 2000 |
| 7 | SURLAG | Surface runoff lag coefficient the running time of soil flow | −0.8 | 0.8 | 0.5 |
| 8 | LAT_TTIME | Surface runoff lag coefficient the running time of soil flow | 0 | 100 | 5 |

(2)　Calibration and validation results

According to the Sufi-2 algorithm in SWAT-cup tool, the output results of the SWAT model were analyzed, calibrated, and verified. In the Sufi-2 algorithm, the P and R factors are used to quantitatively evaluate the uncertainty of the model. The P factor indicates that the measured data fall into the confidence interval of 95% prediction uncertainty (ppu) of the simulation results, and the R factor is the average thickness of the 95 ppu band divided by the standard deviation of the monitoring data. Theoretically, the value of the P factor is between 0 and 100%, while the value of the R factor is between 0 and infinity. The closer the P factor is to 1 and the closer the R factor is to 0, the better the simulation effect. $R^2$ and the Nash–Sutcliffe efficiency coefficient (ENS) were used to comprehensively evaluate the simulation effect of the SWAT model. $R^2$ indicates the consistency of the change trend between the simulated and measured values. The closer the value is to 1, the more

consistent the trend between the simulated and measured values is. The ENS efficiency coefficient indicates the deviation between the measured and simulated values. The closer the value is to 1, the closer the simulated value is to the measured value. According to the geographical location of the subwatershed and hydrological station divided in the early stage, the subwatershed of Yangxian county hydrological station was no. 12, and the subwatershed of Huaxian county was no. 37.

For the calibration and validation of the simulation values of the upper reaches of Hanjiang River, the measured values of Ankang station were used to verify. The measured data of Ankang station were from 1999 to 2015. Therefore, the monthly runoff data from 1999 to 2010 were taken to be the rate period, and the monthly runoff data from 2011 to 2015 were taken to be the validation period. In addition, 1997–1998 was taken to be the warm-up period. Figure S1 in the Supplementary Material shows that the $R^2$ and Nash coefficients were above 0.6. From the calibration validation results, the simulation results were more accurate (see Figure S1 in the Supplementary Material).

In the Weihe River basin, the measured runoff data of Huaxian station in Weihe River in 2020 were used for verification. The rate was regular from day 1 to 60, and the verification period was from 61 to 365 days. Part of the daily runoff data in 2019 were used as the warm-up period. From the validation results, except for the low $R^2$ value in the validation period, the other coefficients were above 0.6 (see Figure S2 in the Supplementary Material). Therefore, the SWAT model of the Weihe River basin in Shaanxi also passed the validation.

On the basis of the results of Figures S1 and S2 (see Supplementary Material), the parameters of Hanjiang River and Weihe River basins in Shaanxi province passed the test, but the parameter values of the Hanjiang River basin rate and validation period were closer to the standard. Therefore, a comparison shows that the applicability of the SWAT model to the Hanjiang River basin was better than that to the Weihe River basin.

### 3.2. Prediction and Simulation of Land-Use Changes Based on the PLUS Model

The PLUS model was used to predict land-use/-cover changes in the Qinling Mountains. The data of influencing factors include the DEM, slope data, distance to the transportation network, distance to residential areas, GDP spatial distribution data, and population spatial distribution data. After buffer-zone analysis, projection conversion, and other processing, the driving factor dataset was obtained (Figure 5).

Using 2020 as the validation year, by comparing actual land-use type data and simulation data in 2020 (Figure 6), and setting the adoption rate to be 0.1, the Kappa coefficient was calculated to be 0.6432, and the overall accuracy was 0.7567.

After model validation, a Markov chain was used to predict land-use types in 2025 and 2030 (Figure 7). Comparing the prediction results with 2020 land-use data (Table 5) shows that the main land-use types in the Qinling Mountains are forestland, agricultural land, and grassland. The reduction in agricultural land is obvious. Forestland increased more, grassland decreased more, and the fluctuation of the water area was large. Construction land decreased first and then increased. Unused land gradually decreased. The main transfer types of forest land were grassland and agricultural land, and the main transfer types of construction land were agricultural land, unused land, and grassland.

**Table 5.** Land-use type area.

|  | Agricultural Land | Forest Land | Grassland | Waters | Construction Land | Unused Land |
|---|---|---|---|---|---|---|
| 2020 | 16,881 | 27,114 | 28,709 | 323 | 524 | 31 |
| 2025 | 11,928 | 57,158 | 3803 | 254 | 439 | 20 |
| 2030 | 11,405 | 57,149 | 3763 | 311 | 959 | 15 |

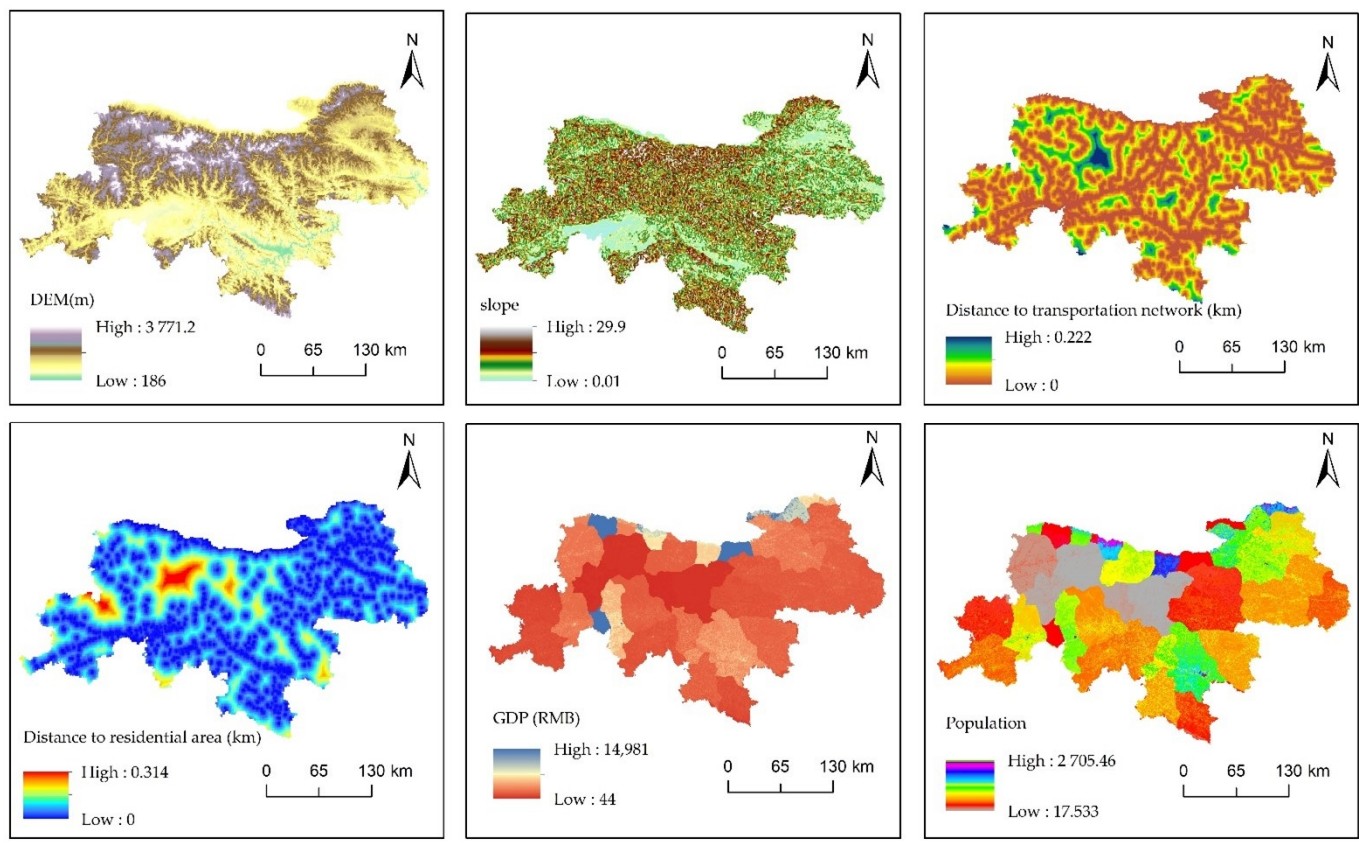

**Figure 5.** Spatial data set of driving factors for land-use type simulation.

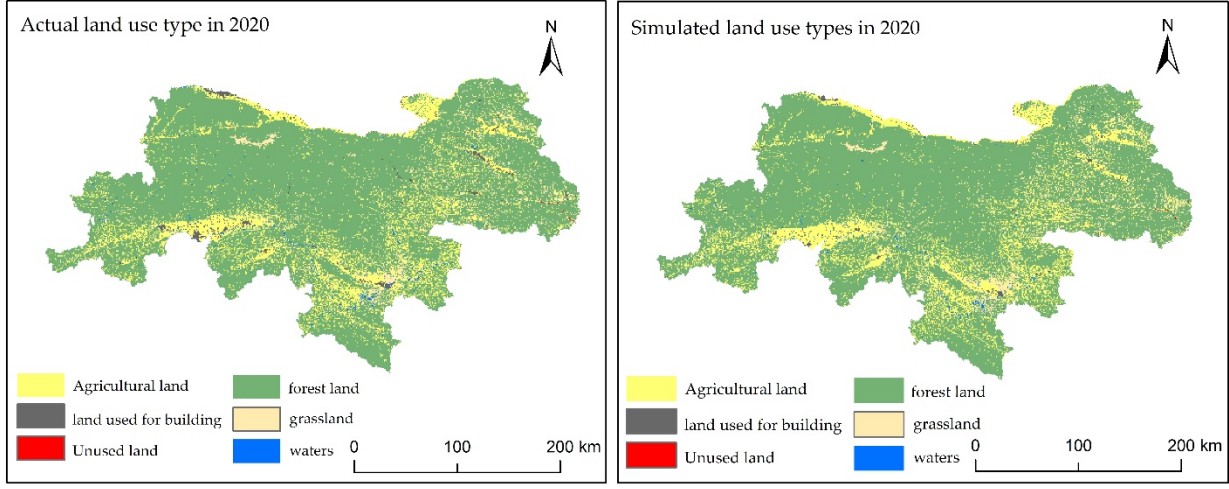

**Figure 6.** Actual land-use and simulation data in 2020.

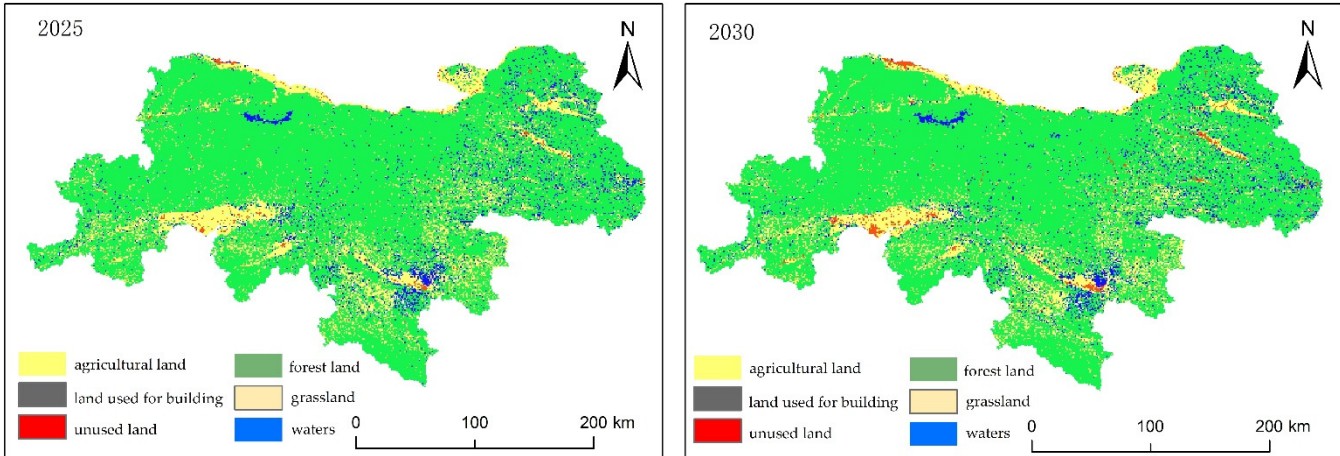

**Figure 7.** Land-use types in 2025 and 2030.

### 3.3. Impact of Land-Use Change on Future Runoff Based on Scenario Analysis

The paper uses future climatic prediction results and a land-use scenario simulation to predict future runoff changes. For the prediction of future climatic data, the annual change trends of temperature and precipitation in the Weihe River and Hanjiang River basins from 2022 to 2035 simulated with the BCC/RCG-WG weather generator are shown in Figure 8. The dot line in the Figure 8 represents the trend of annual average temperature and annual precipitation. The future change trends of temperature and precipitation in the two basins show that the precipitation had an upward trend, and the increase trend of precipitation in the Weihe River was significantly greater than that in the Hanjiang River (Figure 8). From the change trend of temperature, the temperature of Hanjiang River shows an insignificant rise, while the temperature of Wei River shows an insignificant decline. The temperature trend shows that the temperature difference between the two basins is gradually increasing (Figure 8), while the precipitation difference is not obvious.

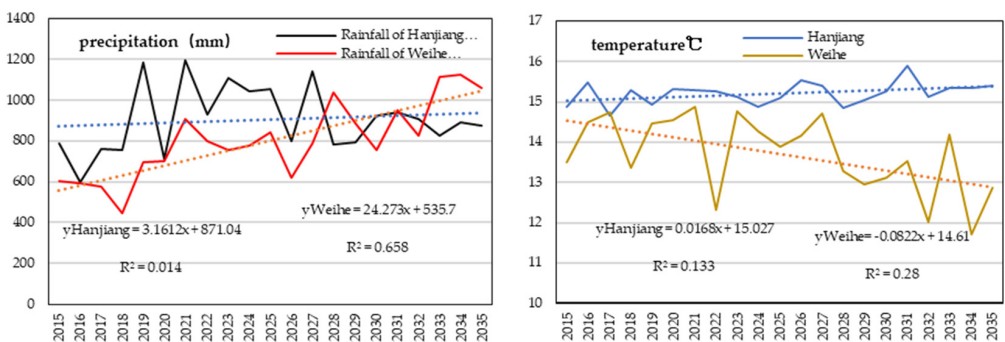

**Figure 8.** Future variation trend of precipitation and temperature in the Qinling Hanjiang River and Weihe River basins.

According to the actual situation of the region, the land-use scenario was set to be the natural-development and forestland-growth scenarios. The natural-development scenario is the natural development of the future land-use type simulated by using the PLUS model after comprehensively considering limiting and influencing factors. The forestland-growth scenario is based on the forestland retention index, which is higher than that in the national forestland protection plan, and a large number of forestland areas exceeding the index are used as the growth quantity of forestland. The two land-use models were input into the successful SWAT model. When simulating 2025, the LUCC in the model was replaced with 2025, and when simulating 2030, the LUCC and meteorological data were replaced with the data of the corresponding year to obtain future runoff changes in the Weihe River and Hanjiang River basins in the two scenarios (Figures 9 and 10).

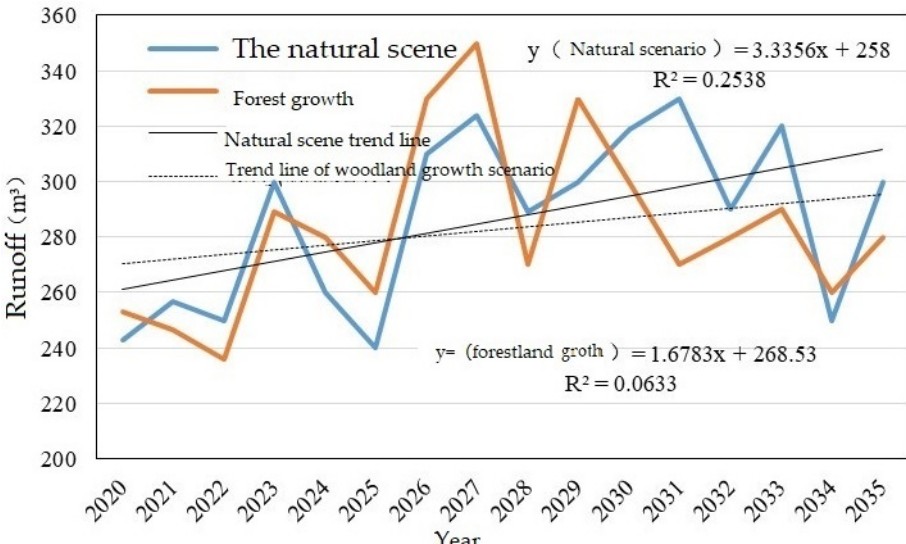

**Figure 9.** Future variation trend of runoff in the Weihe River basin of the Qinling Mountains in different scenarios.

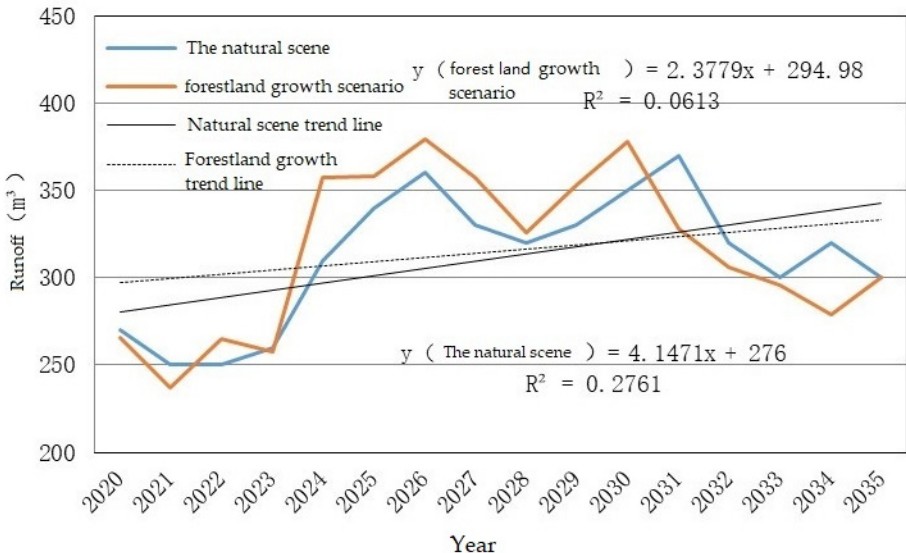

**Figure 10.** Future variation trend of runoff in the Hanjiang River basin of the Qinling Mountains in different scenarios.

The future change trends of the watershed under different scenarios show that the future runoff change of the Qinling Mountains had an increasing trend, while the increase degree of runoff under the natural development scenario was higher than that under the forestland growth scenario, and the increase degree of runoff in the Hanjiang River basin was higher than that in the Weihe River basin as a whole. Therefore, blindly increasing the area of forestland in the Qinling Mountains cannot achieve a better hydrological effect.

## 4. Discussion

The climatic, land-use, and hydrological relationships of the Weihe River basin and the Hanjiang River basin in the Qinling Mountains were constructed through the SWAT model. Model validation results show that SWAT model had a certain level of applicability in both basins. In contrast, the validation results of the Hanjiang River basin were better, simulation accuracy was higher, and applicability was better. For the prediction of future climatic and land-use types, Chinese weather generator BCC/RCG-WG and the PLUS

model, which is the latest land-use simulation model, were used, which are suitable for domestic climatic conditions.

The BCC/RCG-WG model determines the simulation parameters of precipitation and nonprecipitation variables of the model on the basis of the historical meteorological data of daily precipitation, maximal and minimal temperature, and sunshine hours of 672 meteorological stations in China in the last 30 years. With the daily random simulation of various variables in China, the model is more suitable for China, and its prediction results are more in line with the actual situation [48]. Different from related studies, this study uses a suitable weather generator for China to simulate and predict the future climate in the study area, while most studies used the WorldClim database [49] and RCP scenarios [50,51]. Although these climatic simulation results are widely used, they always have problems with data accuracy and the downscaling effect, which affect the accuracy of the results to a certain extent. Therefore, the use of local climatic data for small and medium-sized research in different countries and regions renders the research results more accurate.

The PLUS model is more accurate and convenient for a future land-use model. The data preparation and model test show that the model had good applicability for the simulation of land use in the Qinling Mountains. For the simulation of land use, most other studies used the CA Markov model [52,53]. This paper uses a relatively new land-use simulation model. Compared with the CA Markov model, this model is more advanced and convenient in principle and use, and its results are more accurate [30–32]. Simulation results show that the future transformation trend of land-use types in the Qinling Mountains is an obvious reduction in agricultural land. Forestland would increase more, grassland would decrease more, the fluctuation of water areas would grow, construction land would have an increasing trend, and unused land would gradually decrease. By 2030, the main transfer type of forestland would be grassland, and the main transfer type of construction land would be agricultural land, unused land, and grassland.

The impact of land-use-/-cover (LULC) conditions on global water and energy budgets is critical to understanding climatic intensification in the 21st century [54]. Therefore, in recent years, there have been relatively many studies on the response of land-use and climatic changes to runoff. The research areas are also very extensive, including Serrado in Brazil [55], the Nacqu River basin in the Qinghai Tibet Plateau [56], the northwestern region of São Paulo in Brazil [57], the Khalil River basin in Iran [58], and central Chile [59]. Most of the used methods were SWAT models [60,61], including InVEST [50], AHP [62], VADEQ's VA hydrological model [63], and the DTVGM-CASACNP coupling model [45]. The SWAT model also had good applicability in the Luohe River basin in China. Its future land-use change trend would also increase runoff, and the impact of land-use and climatic changes on runoff shows nonlinear synergy [44]. In addition, the runoff of some areas would show a future decreasing trend, such as the upper reaches of Minjiang River in Southwest China [64] and the Taoer River basin in Northeast China [51]. The research results of the Jinsha River basin in China show that precipitation and temperature in this region have increased significantly, but the upward trend of runoff is not obvious [53]. In addition, relevant research results in Africa show that a certain degree of cultivated-land expansion increases surface runoff [52], and the impact of land-use change is more obvious in arid areas such as in Africa. Different regions in the source area of the Yellow River in China have different factors affecting runoff [65]. In the Lhasa River basin, the impact of land-use change is small, and the increase in temperature increases runoff [66]. In accordance with Chilean regulations, Rebeca Martinez Retureta et al. also considered forest expansion scenarios, and assessed changes in the Quino and Muco River basins in south–central Chile in recent decades. The study designed five scenarios to study the individual and combined effects of LUCC and climate change [67]. Therefore, the response relationship among climate change, land use, and surface runoff had obvious regional differences. Although climate change is global, the change factors of surface runoff are very complex, so the promotion and exploration of relevant research are an important basis for clarifying the influencing factors of global surface runoff. This paper was also

devoted to this aspect of research, but its advancement is reflected in the adoption of the latest land-use and climatic simulation models in accordance with the actual situation in China, and the actual ecological protection policy of China was also considered in the future scenario analysis. These methods render the results of the study more realistic, and can better serve the formulation of environmental policy. Future climatic simulation results and the land-use scenario model were adopted, which fits the actual policy. The results show that using the natural development scenario of land use is highly efficient for future runoff growth, and the excessive completion of forestland growth cannot positively impact the watershed runoff. Therefore, the number of forestland growth areas should not be blindly used as the assessment index in the regional ecological assessment, and the regional ecological performance assessment indicators should be scientifically demonstrated.

A limitation of this study is that it only considered the impacts of climate change and land use on hydrological runoff in the watershed while ignoring the impact of vegetation disturbance and vegetation life history. Regarding this problem, the effect of vegetation disturbance can be ignored because vegetation protection in the Qinling Mountains is perfect due to the vigorous implementation of national environmental protection policies. In the Qinling Mountains, the Chinese government implemented the mountain- and forest-chief systems, which mean that each forest has a specific person in charge. Vegetation growth in the Qinling Mountains shows a good trend year by year. Therefore, in the Qinling Mountains, few major vegetation disturbances occur. In addition, compared with climatic and land-use changes, vegetation growth changes are relatively slower. This study focuses on the impact of climatic and land-use changes on hydrological runoff, and the impact of its life history is more microscopic. Therefore, the impact of vegetation life history can be ignored.

## 5. Conclusions

In this paper, SWAT model was used to construct the response process among climate, land use, and runoff in the Qinling Weihe River and Hanjiang River basins, and the impact of land-use change on runoff under future climate change was verified with realistic scenario analysis. The following conclusions were drawn:

(1) The method adopted in this paper has been widely used in many other regions, but there are differences in climate change and land-use prediction methods. Most studies use international climate scenario data for climate change prediction, and the downscaling process often affects the simulation results. The land-use and land-cover change (LUCC) scenario settings of most studies were not analyzed in combination with national policies and the specific conditions of the study area. These problems can affect the results of research and the analysis of scientific problems. On the basis of the coupling of the new land-use model (PLUS), a land-use scenario was established by combining actual local policies, and the domestic weather generator model was used in climate change prediction. This is more realistic and enhances the practicability of the research results, so as to better serve policy making.

(2) The protection of the Qinling forest area should follow the scientific method, and the evaluation index of ecological achievements should also pass scientific demonstration. Most of the previous studies focused on simple climate change and vegetation response, and most of the research results did not involve policy services. Translating research results into usable policies is the main direction of scientific research, and this direction is the main scientific problem of future climatic and land-use change research.

(3) The hydrological model was first used to simulate the response relationship among climate, land use, and runoff in the northern and southern Qinling valleys. The results also reflect the north and south Qinling mountains' differences in climate, land-use, and runoff changes. As reference for related research, the results can also be used as a scientific reference for ecological-protection and land-use planning policies in

order to realize sustainability and economic development in the Qinling ecological environment while mitigating the influence of climate change.

**Supplementary Materials:** The following supporting information can be downloaded at: https://www.mdpi.com/article/10.3390/f13111776/s1, Figure S1: The calibration and validation effect of SWAT simulation results in Shaanxi Hanjiang River Basin; Figure S2: The calibration and validation effect of SWAT simulation results in Weihe River Basin in Shaanxi Province.

**Author Contributions:** Conceptualization, X.M. and K.Z.; methodology, X.M. and J.L.; software, X.M. and J.L.; validation, X.M., J.L. and C.D.; formal analysis, X.M.; investigation, X.M.; resources, X.M. and K.Z.; data curation, X.M. and K.Z.; writing—original draft preparation, X.M.; writing—review and editing, X.M.; visualization, X.M.; supervision, K.Z.; project administration, K.Z.; funding acquisition, K.Z. All authors have read and agreed to the published version of the manuscript.

**Funding:** This research was funded by Luo H., Shaanxi Key Industry Innovation Chain (Group) project, grant number 2020ZDLSF06-02; Xiao S., The National Natural Science Foundation of China, grant number 42171158; Ma X.P., the Natural Science Basic Research Project of Science and Technology Department of Shaanxi Province (Youth Project), grant number 2021JQ-818; Li J., the General Project of National Natural Science Foundation of China, grant numbers 42071285, 41771576. The APC was funded by Zhao K.F. and Luo H., Shaanxi Key Industry Innovation Chain (Group) project, grant number 2020ZDLSF06-02.

**Data Availability Statement:** The data are not publicly available as they will be used in future studies.

**Conflicts of Interest:** The authors declare no conflict of interest.

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
