# Peer review of "The Effects of Land-Use and Climatic Changes on the Hydrological Environment in the Qinling Mountains of Shaanxi Province"

_forests, doi:10.3390/f13111776_

Round 1

Reviewer 1 Report

The manuscript entitled “Study on the response relationship between land use and climate change and its hydrological environment effect in Qinling mountains of Shaanxi Province” by Zhao et al. conducted very interesting and important research in China. Currently, there is a critical need in addressing the effects of land cover and land use and climate changes on water resources in the Qinling Mountains. Personally, I am so pleased to see this kind of research in this region. However, this manuscript is hard to understand due to poor writing. Many descriptions seem directly translated by some software.  In addition, one of the limitations of this research design is that there is only one-year of hydrology data used for calibration and validation, which is not reliable and partly explained why validation efficiency values were low. Therefore, my recommendation for this manuscript is major revision. Authors should work hard to improve the quality of the manuscript.

I also provide some specific comments which may help your revision.

1.      Title can be rephrased to “The effects of land use and climate changes on the hydrological environment in Qinling Mountains of Shaanxi Province”.    

2.      In this manuscript, you used tooooo many semicolons, which should be avoided in your future writing. In addition, this is the basic grammatical rule. Moreover, there is no “” in English and authors should know this common perception.

3.      In your introduction, you talked too much about the importance of Qinling Mountains and there is no logic between paragraphs. The first paragraph is ok. In your second paragraph, you should discuss the preliminary research needs in this region. The third paragraph you can introduce SWAT (please remember, SWAT should be all capital letters) and PLUS. Finally, you can lay out your research objectives.

4.      Please remember that in your method section, you can only describe the method you used. However, you put a large amount of model build as your results. Please move them to the method section. Model building is the method, no matter how many HRUs and subwatershed were divided.  

5.      In model calibration, you only have one year data to calibrate and validate your model. In the model, there is a long period of data is needed to warm up your model. You can repeat your one-year data ten times. Then, use first five years for model calibration and last five-years for model validation.

6.      The writing should be significantly improved before you submit it to any journals. The present manuscript can not be accepted by any journals.  

Author Response

We would like to thank the review experts for their suggestions on this paper, which we believe are very helpful for the improvement of the paper. We have made corresponding modifications to the paper based on the experts' opinions, and the modification instructions are as follows:

General Comments: The manuscript entitled: Study on the response relationship between land use and climate change and its hydrological environment effect in Qinling mountains of Shaanxi Province is a research about the water resources response to land use and climate change and can be applied to worldwide. The objective was achieved according to the results. The objective was achieved according to the results. The methodology is timely but not is well presented. The manuscript needs to improve in writing and English. Therefore, it can be accepted for publication after clarifying some important comments.

  1. I suggest to the authors remove in the title Study on the and star with Response… if they consider it.

Modified.

  1. The summary can be improved with some statistical results.

The issue has been fixed.

The temperature in the Qinling mountains is mainly concentrated at 10-15℃. From 2010 to 2020, the area occupied by 10-15℃ increased by 22.44%. The precipitation is mainly over 1000mm. From 2010 to 2020, the area of precipitation over 1000mm increased by 12.71%. The land use type most affected by climate change is forest land, followed by agricultural land and grassland. The SWAT model is applicable to the Qinling Valley, and the simulation results are verified, and the Nash coefficient is above 0.6. Under the future climate change and land use patterns, the runoff in the Qinling mountains watershed showed an upward trend, and the runoff in the Hanjiang River basin increased more than that in the Weihe River Basin, with a change rate of 47.471m³/10a and 33.356m³/10a respectively. According to the future trend of the two different scenarios, the increase degree of runoff in the natural scenario of Weihe River basin was 16.567 m³/10a higher than that in the woodland scenario, and the increase degree of runoff in the Hanjiang River basin was 17.692 m³/10a higher than that in the woodland scenario. Therefore, blindly increasing the forestland area in Qinling mountains can not achieve better hydrological effect.

  1. It is suggested to include the word statistically or statistics when referring to the trend of temperatures or rainfall throughout the manuscript. It is important to categorize that trend, is it significant or not significant from the statistical point of view. Example: statistically significant increase in temperature or statistically significant decrease trend.

Modified.

  1. First sentences or affirmation need reference.

Modified.

  1. Be careful with some commas, periods, and semicolons throughout the manuscript.

Modified.

  1. The English should be checked.

Modified.

  1. Figures in all manuscript need better resolution.

Modified.

  1. Figure 1 can be improved with an international location context of China and Shaanxi Province. The digital elevation model could be included to distinguish the Qinling Mountains.

Modified.

  1. Where are the meteorological stations located? Can be present in Figure 1.

Modified.

  1. Line 166: maximum / minimum temperature? or daily maximum / minimum temperature.

Modified.

  1. 90m × 90m DEM from China geospatial data cloud was used, but what was the web site? Why not use another DEM with better resolution?

The source website of DEM data has been added in the revised draft. The website is http://www.gscloud.cn/

DEM with a resolution of 90m has a better effect in delineating the molecular watershed. If the accuracy is too high, the model will collapse or the calculation of sub-watershed will be redundant, which will affect the simulation results to a certain extent. For relevant research results, please refer to the literature "Scale Effect Analysis of hydrological Process Based on SWAT Model -- Taking the Upper reaches of Han River as an example". This study analyzed the effect of topographic scale on the results of hydrological processes. With the increase of DEM sampling grid area, the sampling grid of sediment yield in small watershed shows a downward trend from 30m to 1020m, which is highly correlated with the average slope and controllable. Both sediment yield and runoff in the upper reaches of the Han River show a downward trend when they fluctuate. When the grid is at the level of 30m to 270m, the influence of grain size change on hydrological process has obvious regularity, which can be summarized by using certain mathematical models, and the DEM resolution for the study of hydrological process can be increased by 270m.

  1. What is the spatial resolution of the LUCC data? How was obtained it? (classification method and web site?

Land use/land cover change (LUCC) data were obtained from the Data Center for Resources and Environmental Sciences, Chinese Academy of Sciences (http://www.dsac.cn/) at a resolution of 30m. Landsat TM/ETM/OLI remote sensing images were used as the main data source. Through image fusion, geometric correction, image enhancement and Mosaic, the human-computer interaction visual interpretation method was used to classify the land use types in China into six first-level categories, including forest land, grassland, water source, residential land and unused land.

The problem has been corrected in the text.

  1. The methodology does not show information about the time period of the SWAT model warm-up, calibration and validation process.

In the Hanjiang River basin, the measured runoff data from Ankang Station are used to verify the simulation results. The monthly runoff data from 1997 to 1998 are taken as the warm-up period, the monthly runoff data from 1999 to 2010 are taken as the rate period, and the period from 2011 to 2015 is taken as the verification period. For details, see Figure 8 of the revised draft and the calibration verification section.

In the Weihe River Basin, part of the daily runoff data in 2019 was used as the warm-up period, the daily runoff data from the 1st to the 60th day of 2020 was used as the rate period, and the daily runoff data from the 61-365 days was used as the validation period. For details, see Figure 9 of the revised version and the calibration and verification section.

  1. The calibration and validation of the PLUS model is missing in methodology.

PLUS model is used here to simulate future land use. The verification of PLUS simulation results in this paper is shown in Figure 12. Firstly, land use data in 2010 and 2015 are used to simulate land use in 2020, and the simulated results are compared with actual land use data to obtain Kappa coefficient 0.7567. It shows that the model can be used to predict the land use data of future years through testing.

  1. The presentation of results should be improved.

Modified.

  1. Was the slope not calculated to run the SWAT model?

During the SWAT model run, the input data includes the DEM, and the model automatically calculates the slope during the HRU generation stage.

  1. Line 332: The spatial resolution of LUCC is need present in methodology section.

Modified.

  1. Lines 324-336: It can be eliminated or summarized because this information is in the methodology, it is not necessary to present it again.

Modified.

  1. Table 6 is better in methodology.

Modified.

  1. What method was used to calculate evapotranspiration?

Evapotranspiration includes evaporation, transpiration and sublimation of canopy trapped water and evaporation of soil water. Evapotranspiration is the main route by which water is transferred out of river basins. In many river basins and continents except Antarctica, evaporation is greater than runoff. Accurate evaluation of evapotranspiration is the key to estimating water resources, and it is also the key issue to study the impact of climate and land cover change on river runoff.

(1) Potential evapotranspiration

The model provides Penman-Monteith, Priestley-Taylor and Hargreaves as three methods to calculate potential evapotranspiration. In addition, measured data or calculated daily potential evapotranspiration data can be used. The Penman-Monteith formula requires input data for radiation, air temperature, wind speed, and relative humidity (these data are prepared as required before the model is run).

(2) Actual evapotranspiration

Actual evapotranspiration is calculated on the basis of potential evapotranspiration. In the SWAT model, the evaporation trapped by the vegetation canopy is calculated first, then the maximum transpiration, maximum sublimation and maximum soil water evaporation are calculated, and finally the actual sublimation and soil water evaporation are calculated.

The canopy traps evaporation. When calculating the actual evaporation, the model assumes that the water trapped by the canopy should be evaporated as much as possible. Two cases are considered: (1) The potential evaporation is less than the free water trapped by the canopy; (2) Potential evaporation is greater than the content of free water trapped by the canopy; When all the free water trapped by the vegetation canopy is evaporated, the water needed for further evaporation is obtained from the vegetation and the soil.

Plants transpiration. Assuming that vegetation grows under an ideal condition, the calculation formula is:

Where, Et is the maximum transpiration on a certain day, mm; E'0 is the potential evaporation adjusted by free water evaporation in vegetation canopy, mm; LAI is leaf area index. The calculated transpiration may be somewhat larger than the actual transpiration.

Soil water evaporates. In the calculation of soil water evaporation, the evaporation required by different depth soil layers is distinguished first, and the division of soil depth levels determines the maximum evaporation allowed by soil. The amount of water needed for soil water evaporation is determined by the evaporation requirement of upper soil and lower soil.

It is shown above that for the division of soil depth, it is assumed that 50% of the evaporation water requirement is provided by the water content of the upper soil within 0-10 mm. Therefore, 50 mm of the evaporation water requirement of 100 mm is provided by the upper soil, which obviously cannot meet the demand. Therefore, The SWAT model establishes a coefficient to adjust the division of soil depth to meet the evaporative water demand. The adjusted formula can be expressed as follows.

Where, esco is the regulation coefficient of soil evaporation, which is proposed by SWAT model to adjust the effects of soil capillary action and soil cracks on evaporation in different soil layers. Different esco values correspond to the corresponding depth of soil layer division.

As ESCO values decrease, the model is able to obtain water supply evaporation from deeper soil. When the soil water content is lower than the field water capacity, the evaporation water requirement also decreases.

  1. It is recommended to change the word verification by validation.

Modified.

  1. The discussion can be improved with some works that show results and themes similar to this manuscript but applied to other regions and climates. To suggest some I mention the following:

它已经过修订。非常感谢审稿人提供的三篇参考文献,这些参考文献与本文的研究非常相关。我们在讨论中增加了这三个参考文献,这也使本文的讨论更加丰富和深入。

  1. 可以通过强调该方法在其他区域的应用来总结和改进这一结论。

改 性。

Reviewer 2 Report

Study on the response relationship between land use and cli-mate change and its hydrological environment effect in Qinling mountains of Shaanxi Province

General Comments: The manuscript entitled: Study on the response relationship between land use and cli-mate change and its hydrological environment effect in Qinling mountains of Shaanxi Province is a research about the water resources response to land use and climate change and can be applied to worldwide. The objective was achieved according to the results. The objective was achieved according to the results. The methodology is timely but not is well presented. The manuscript needs to improve in writing and English. Therefore, it can be accepted for publication after clarifying some important comments.

1. I suggest to the authors remove in the title Study on the and star with Response… if they consider it.

2. The summary can be improved with some statistical results.

3. It is suggested to include the word statistically or statistics when referring to the trend of temperatures or rainfall throughout the manuscript. It is important to categorize that trend, is it significant or not significant from the statistical point of view. Example: statistically significant increase in temperature or statistically significant decrease trend.

4. First sentences or affirmation need reference.

5. Be careful with some commas, periods, and semicolons throughout the manuscript.

6. The English should be checked.

7. Figures in all manuscript need better resolution.

8. Figure 1 can be improved with an international location context of China and Shaanxi Province. The digital elevation model could be included to distinguish the Qinling Mountains.

9. Where are the meteorological stations located? Can be present in Figure 1.

10. Line 166: maximum / minimum temperature? or daily maximum / minimum temperature.

11.  90m × 90m DEM from China geospatial data cloud was used, but what was the web site? Why not use another DEM with better resolution?

12. What is the spatial resolution of the LUCC data? How was obtained it? (classification method and web site?

13. The methodology does not show information about the time period of the SWAT model warm-up, calibration and validation process.

14. The calibration and validation of the PLUS model is missing in methodology.

15. The presentation of results should be improved.

16. Was the slope not calculated to run the SWAT model?

17. Line 332: The spatial resolution of LUCC is need present in methodology section.

18. Lines 324-336: It can be eliminated or summarized because this information is in the methodology, it is not necessary to present it again.

19. Table 6 is better in methodology.

20. What method was used to calculate evapotranspiration?

21.  It is recommended to change the word verification by validation.

22. The discussion can be improved with some works that show results and themes similar to this manuscript but applied to other regions and climates. To suggest some I mention the following:

https://doi.org/10.3390/w13060794 ; https://doi.org/10.1016/j.jhydrol.2021.126047;  https://doi.org/10.3390/w14152304

23.  The conclusion can be summarized and improved by highlighting the application of the methodology in other regions.

Author Response

The manuscript entitled “Study on the response relationship between land use and climate change and its hydrological environment effect in Qinling mountains of Shaanxi Province” by Zhao et al. conducted very interesting and important research in China. Currently, there is a critical need in addressing the effects of land cover and land use and climate changes on water resources in the Qinling Mountains. Personally, I am so pleased to see this kind of research in this region. However, this manuscript is hard to understand due to poor writing. Many descriptions seem directly translated by some software.  In addition, one of the limitations of this research design is that there is only one-year of hydrology data used for calibration and validation, which is not reliable and partly explained why validation efficiency values were low. Therefore, my recommendation for this manuscript is major revision. Authors should work hard to improve the quality of the manuscript.

I also provide some specific comments which may help your revision.

  1. Title can be rephrased to “The effects of land use and climate changes on the hydrological environment in Qinling Mountains of Shaanxi Province”.    

Modified.

  1. In this manuscript, you used too many semicolons, which should be avoided in your future writing. In addition, this is the basic grammatical rule. Moreover, there is no “” in English and authors should know this common perception.

Modified.

  1. In your introduction, you talked too much about the importance of Qinling Mountains and there is no logic between paragraphs. The first paragraph is ok. In your second paragraph, you should discuss the preliminary research needs in this region. The third paragraph you can introduce SWAT (please remember, SWAT should be all capital letters) and PLUS. Finally, you can lay out your research objectives.

Modified.

  1. Please remember that in your method section, you can only describe the method you used. However, you put a large amount of model build as your results. Please move them to the method section. Model building is the method, no matter how many HRUs and subwatershed were divided.  

      Modified.This section has been adapted into the method.

  1. In model calibration, you only have one year data to calibrate and validate your model. In the model, there is a long period of data is needed to warm up your model. You can repeat your one-year data ten times. Then, use first five years for model calibration and last five-years for model validation.

       In this paper, only the daily runoff data of 2020 were used to verify the SWAT model of the Weihe River Basin, mainly because only the measured daily runoff data of 2020 and part of the daily runoff data of 2019 were obtained. Therefore, part of the daily data in 2019 is taken as the warm-up period, 1-60 days in 2020 as the rate period, and 61-365 days as the verification period. If the rate periodic and verification period are defined according to the year, it is suitable to use monthly data, but the monthly data are relatively small, and if repeated for 10 years, the verification results will be inaccurate. Therefore, in this paper, the daily data of one year is used for calibration verification in the Weihe River Basin, which can ensure the accuracy of the results.

  1. 在你把它提交给任何期刊之前,写作应该得到显着的改进。本稿件不能被任何期刊接受。

改 性。

Round 2

Reviewer 1 Report

The manuscript has made a substantial improvement in the revised version. The current version still has some grammatical mistakes and typos. Therefore, the manuscript should carefully address the description.

Here are some examples for your references.

1.      Line 11 please change hydrological effects, to hydrological responses

2.      Line 11, delete the response in “response relationship” and others throughout the manuscript.

3.      Line 15 in order to find

4.      Line 83, SWAT is a semi-distributed model

5.      In tables, forestland should be two words, i.e. forest land

6.      In this manuscript, the authors have provided very detailed expression of the data sources and calibrations. I suggest can move some materials into supplementary materials.

Reviewer 2 Report

Dear authors, Thank you very much for considering my suggestions.

Author Response

Many thanks for the reviewer's advice.